# Overconfidence, Financial Advice Seeking and Household Portfolio Under-Diversification

**Stijn P. M. Broekema** [1,2] **and Marc M. Kramer** [1,]*

1    Faculty of Economics & Business, University of Groningen, Nettelbosje 2, 9747 AE Groningen,
     The Netherlands; stijnbroekema@gmail.com
2    Independent Researcher, 1017 AS Amsterdam, The Netherlands
*    Correspondence: m.m.kramer@rug.nl

**Abstract:** This paper examines the relationship between overconfidence and losses from under-diversification among Dutch investors. We find that a lack of proper portfolio diversification is positively associated with overconfidence. Part of this relationship is mediated through the lower propensity of overconfident individuals to hire a professional financial adviser. We use data from the 2005 wave of the DNB Dutch Household Survey that provides us with detailed portfolio data of 257 investors. We proxy for overconfidence by exploiting the difference between measured and self-assessed financial literacy, and use this proxy in a regression model (with and without mediation) to explain the difference between the actual households return and the return that could have been obtained by selecting a portfolio on the efficient frontier with equivalent risk. Our results contribute to the current discussion among policy makers on the role of financial advice and self-perceptions in household financial decision-making.

**Keywords:** portfolio diversification; overconfidence; financial advice; financial literacy; individual investors



## 1. Introduction

Lack of proper portfolio diversification is one of the most prevalent and costly investment mistakes that households make (Gomes et al. 2021). Under-diversification refers to a portfolio that provides a subpar expected return given the amount of risk taken. Statman (1987) indicates that a well-diversified portfolio consists of at least thirty stocks, while Goetzmann and Kumar (2008) find that portfolios of most individual investors contain a maximum of five stocks. Compared to the most diversified investors, the least diversified investors in their study earn an annual risk-adjusted return that is on average 2% lower.

An unresolved question is why so many investors hold underdiversified portfolios. Although transaction costs and a lack of sophistication may explain some part of the puzzle, most agree that one need also needs to rely on behavioral biases, most notably overconfidence, to explain the phenomenon fully.

Overconfidence has been associated with various suboptimal financial behaviors. Grinblatt and Keloharju (2009) and Barber and Odean (2000) for example, provide evidence of overconfidence driving excessive trading, while Cox and Zwinkels (2016) relate overconfidence to a lack of proper mortgage insurance. Some authors already provide indicative evidence of overconfidence driving a lack of proper diversification. Goetzmann and Kumar (2008) report that the degree of under-diversification correlates with other investor behaviors that link to overconfidence. In their sample, investors that trade excessively exhibit also a greater degree of under-diversification. Døskeland and Hvide (2011) relate retail investor underperformance to the overweighting of professionally close stocks and argue this may result from overconfidence. Von Gaudecker (2015) indicates that low literate investors that do not obtain financial advice incur most diversification losses and

concludes that overconfidence is a plausible interpretation of that result. Our paper will formally test that interpretation, and is to our knowledge, the first to do so.

Overconfidence is a widespread behavioral phenomenon, which relates to various concepts. The miscalibration type of overconfidence refers to the overestimation of the precision of one's knowledge reflected in setting too narrow confidence intervals in knowledge questions. Well-known in this domain is miscalibration among CFO's when estimating next year's stock market returns (Boutros et al. 2020). The better than average effect refers to overestimating one's position in the distribution of some positive dimension within a group (Svenson 1981). Illusion of control refers to the perception to be able to influence random events (Langer 1975). Biased self-attribution, the unwarranted credit for one's past successes, in turn may drive overconfidence.

In theory, two distinct mechanisms may link overconfidence to under-diversification. First, because overconfident investors simultaneously underestimate the variance of stock returns (Daniel and Hirshleifer 2015) and overestimate their ability to pick the right stocks (Barber and Odean 2002), it makes sense to hold a focused portfolio with a lot of idiosyncratic risk. If these beliefs are incorrect, however—as implied by overconfidence—these investors lose money. Second, by overstating their own knowledge, overconfident investors see less need to ask for assistance in their investment decision-making. Indeed, Gentile et al. (2016) find that Italian investors that overestimate their financial knowledge exhibit a significant lower propensity to rely on professional financial advice, while Kramer (2016) finds that especially most confident investors ask for advice less.

Various studies indicate a rather negative impact of financial advisors on the quality of financial choices (Chalmers and Reuter 2020; Hackethal et al. 2012). This effect has mostly been attributed to conflicts of interest and commission-based pay of most advisors. Other studies indicate that investors that do hire a professional financial advisor improve portfolio diversification (Hackethal et al. 2012).

We thus test for both a direct link between overconfidence and under-diversification, and an indirect (or mediated) effect that runs through the propensity to choose for assistance by a professional financial expert. We measure the degree of under-diversification applying the diversification loss introduced by Calvet et al. (2007).

Based on the short discussion above, we expect to find that: (H1) overconfidence has a positive effect on diversification losses, (H2) professional financial advice has a negative effect on diversification losses, (H3) overconfidence has a negative effect on the propensity to seek professional financial advice, and (H4) financial advice (partially) mediates the relationship between overconfidence and investment performance.

This paper contributes to the literature in various ways. First, we use a direct measure for overconfidence by exploiting the difference between perceived and actual financial literacy and link this to an objective measure of the reduction in portfolio returns that results from under-diversification. Second, by using a mediation analysis we aim to identify both the direct effect as well as the mediating role of financial advice in this relationship. Third, by including both common stocks, and other portfolio assets such as mutual funds, we provide a more accurate picture of under-diversification. Calvet et al. (2007) show that by looking at individual stocks only, as is performed in many related studies, one may actually overestimate the under-diversification problem.

Using data from the DNB Household Survey (DHS), we show that most people appear to incur modest losses from under-diversification. Yet, overconfident investors incur significantly higher losses. The quartile of most overconfident households incurs a return loss that is almost 70% higher than the quartile of people that score low on our overconfidence measure. Most of this loss occurs through the direct channel, while the mediating effect of financial advice accounts is small, but significant.

Our results contribute to the current discussion among policymakers that aim to improve household financial decision-making. It appears that professional financial advice relates to better portfolio decision-making when it comes to diversification. While we

cannot establish causal links given the nature of our data set, our results warrant more study to the impact of financial advice taking, and the role of biased self-perceptions.

## 2. Data and Methods

### 2.1. DNB Dutch Household Survey

We use data from the Dutch Central Bank (DNB) Household Survey (DHS), which is part of the CentER panel. The survey is administered through the Internet. To prevent selection bias, CentER data provides households that do not have Internet with access. The data is considered to be of high quality and a close representation of the Dutch population. It is used frequently in studies on financial consumer decision-making (e.g., Bucher-Koenen et al. 2017).

We use the DHS data for two primary reasons. First, the survey does not only ask respondents to disclose whether they invest in a certain asset class such as stocks, but also asks to list in what specific assets they invest and for what amounts. This yields the opportunity to analyze the risk and return properties of households' portfolios. Second, the data contains a large variety of socio-economic and psychological variables that we use as control variables.

Specifically, we use the 2005 DHS wave. The advantage of this wave is that it includes an extra module designed to extensively measure financial literacy (please refer to Van Rooij et al. 2011 for a detailed overview of the financial literacy questions and their statistics). We exploit the difference between this objective measure of financial literacy and perceived levels, to derive our main proxy for overconfidence.

This extra module was sent out to 2028 households, of which 1508 responded. Households for which no information on their main source of advice is available are excluded from the sample. This yields a sample of 1277 observations, of which 350 are investors (i.e., people who invest in at least one of the following: shares, funds, bonds or options). A part of these observations cannot be used because respondents did not list their portfolio items, or because items that are listed could not be unambiguously matched to a return series. This results in a final sample of 257 household portfolios.

### 2.2. Key Variable Construction

#### 2.2.1. Under-Diversification

Following the paper by Von Gaudecker (2015) and Calvet et al. (2007), we measure the degree of under-diversification by calculating the return loss. It measures how much expected return an investor loses by not choosing an optimal portfolio. It equals the difference between the actual households return and the return that could have been obtained by selecting a portfolio on the efficient frontier with equivalent risk. In other words, it quantifies the economic losses resulting from under-diversification.

More specifically, the return loss is defined as:

$$RL_h = \mu_m \cdot \omega_h \cdot \beta_h \cdot \left( \frac{S_m - S_h}{S_h} \right), \tag{1}$$

where $RL_h$ is the return loss for household $h$, $\mu\_m$ equals the expected excess return of the market portfolio, $\omega_h\_$ represents the weight in risky assets of household $h$, $\beta_h$ refers to the household beta, $S_m\_$ equals the Sharpe ratio of the market portfolio: the excess market return scaled by its standard deviation ($S_m\_ = \mu_m\_/\sigma_m$), and $S_h\_$ represents the Sharpe ratio of the portfolio of household $h$[1].

To derive the return loss, we first match each listed item in a household's portfolio to the item's return series in Eikon-DataStream. We then estimate the household beta by using the vector of weights and the asset betas that we calculate by imposing the CAPM. We also estimate the asset's variance–covariance matrix, from which we infer the standard deviation of the household's portfolio. Then we have all the inputs to calculate the Sharpe ratio's and the household's return loss.

Following Von Gaudecker (2015), we proxy for the risk-free rate with the one-month Euribor rate, and for the efficient market portfolio with the MSCI Europe Index. The results are robust to using the MSCI World Index or Dutch AEX Index instead.

### 2.2.2. Overconfidence

We measure overconfidence in the same vein as Abreu and Mendes (2012), and Grinblatt and Keloharju (2009) and exploit the difference between perceived and measured financial literacy. In fact, we correct someone's self-assessed financial literacy for the part that reflects actual financial literacy, and thus, obtain overconfidence from the residual effect. This operationalization most closely represents the miscalibration manifestation of overconfidence. As a robustness test, we use the respondents' estimates of inflation. Specifically, we create a dummy which equals one for people whose highest and lowest inflation estimates differ by less than one percentage point, and zero otherwise. Although narrow confidence intervals may represent superior capability in inflation forecasting, Mankiw et al. (2003) shows it may actually reflect overconfidence. Given our sample of ordinary households that mostly lack the sophistication to understand drivers of inflation, we believe the latter explanation to be most convincing. To be sure, we compared how well the inflation-range estimates fitted realized inflation. In the group, that we labelled as overconfident based on their inflation expectations, almost 70% was wrong (and thus were surprised by the actual inflation realization), while in the non-overconfident group this was only 44%.

We derive measured financial literacy from the financial literacy module. More specifically, following Van Rooij et al. (2011) we perform a factor analysis using the advanced financial literacy questions (please refer to Van Rooij et al. (2011) for the exact wording of the questions). Specifically, for each of the 11 financial literacy questions, we create a dummy indicating whether the respondent answered the question correctly, and a dummy to indicate whether the respondent chooses the "do not know" option. Van Rooij et al. (2011) indicated the importance of taking the difference between a "do not know" and a wrong answer explicitly into account. We thus carry out a factor analysis on 22 dummies. Bartlett's test of sphericity ($p < 0.01$) and the Kaiser–Meyer–Olkin (KMO) measure of sampling adequacy (KMO = 0.92) indicate that a factor analysis is appropriate. The factor scores are obtained using Bartlett's method (Bartlett 1937).

Self-assessed literacy is using the response on the following question: "How knowledgeable do you consider yourself with respect to financial matters?" People answer this question on a 4-point scale from "not knowledgeable" to "very knowledgeable".

We then regress someone's self-assessed financial literacy on measured financial literacy and obtain the standardized residuals to reflect the degree of overconfidence.

### 2.2.3. Financial Advice

Financial advice is obtained from the following question: "What is your most important source of advice when you have to make important financial decisions for the household?" To operationalize professional financial advice, a dummy is created that is set equal to one if the respondent answered "professional financial adviser", and zero otherwise.

### 2.2.4. Control Variables

With respect to the control variables, we follow the literature and include: age, gender, education, income[2], primary occupation, having a partner, having kids, and total wealth.

### 2.3. Summary Statistics

Table 1 provides an overview of descriptive statistics of both the full DHS sample (Column 1), the subsample of all investors (Column 2), and the subsample of investors for which portfolios could unambiguously be established (Column 3). We observe some noteworthy differences between the full DHS sample and the sample that we use in this

study. Our sample contains more males, is older on average, wealthier, has a higher income, and is more financially literate. It mostly resembles differences between investors and non-investors that have been observed in many other studies. Comparing all investors in the DHS sample (Column 2) and the sample that we use (Column 3), we see that although there are some differences, these are generally small, and unlikely to lead to sample selection bias.

**Table 1.** Descriptive statistics.

| **Full DHS Sample** | | | | | |
|---|---|---|---|---|---|
| | **Mean** | **Med.** | **SD** | **Min** | **Max** |
| Return loss (in base points per year) | | | | | |
| Overconfidence | 0.00 | −0.25 | 1.00 | −1.93 | 3.26 |
| Professional financial adviser (%) | 27.10 | | | | |
| Measured financial literacy (# correct questions) | 6.40 | 6.00 | 2.95 | 0.00 | 11.00 |
| Self-assessed literacy (in range 1–4) | 2.15 | 2.00 | 0.69 | 1.00 | 4.00 |
| Age (yrs) | 50.30 | 50.00 | 15.00 | 22.00 | 90.00 |
| Wealth (x €1000) | 161.04 | | 188.39 | | |
| Gross income (x €1000) | 29.91 | | 22.86 | | |
| Male (%) | 57.00 | | | | |
| Higher vocational or university education (%) | 39.20 | | | | |
| Employee (%) | 50.80 | | | | |
| Partner (%) | 67.70 | | | | |
| Households with kids (%) | 34.80 | | | | |
| N | 1.277 | | | | |

| **Full DHS Sample [Investors Only]** | | | | | |
|---|---|---|---|---|---|
| | **Mean** | **Med.** | **SD** | **Min** | **Max** |
| Return loss (in base points per year) | | | | | |
| Overconfidence | 0.19 | −0.27 | 1.00 | −1.93 | 2.76 |
| Professional financial adviser (%) | 27.70 | | | | |
| Measured financial literacy (# correct questions) | 8.20 | 8.00 | 2.36 | 0.00 | 11.00 |
| Self-assessed literacy (in range 1–4) | 2.35 | 2.00 | 0.69 | 1.00 | 4.00 |
| Age (yrs) | 53.70 | 54.00 | 14.30 | 24.00 | 90.00 |
| Wealth (x €1000) | 257.06 | | 239.82 | | |
| Gross income (x €1000) | 37.25 | | 25.03 | | |
| Male (%) | 71.10 | | | | |
| Higher vocational or university education (%) | 52.30 | | | | |
| Employee (%) | 50.60 | | | | |
| Partner (%) | 74.30 | | | | |
| Households with kids (%) | 28.40 | | | | |
| N | 350 | | | | |

| **Sample Used in This Study** | | | | | |
|---|---|---|---|---|---|
| | **Mean** | **Med.** | **SD** | **Min** | **Max** |
| Return loss (in base points per year) | 0.57 | 0.28 | 0.99 | 0.01 | 8.77 |
| Overconfidence | 0.23 | −0.28 | 0.96 | −1.93 | 2.76 |
| Professional financial adviser (%) | 24.10 | | | | |
| Measured financial literacy (# correct questions) | 8.60 | 8.00 | 2.25 | 0.00 | 11.00 |
| Self-assessed literacy (in range 1–4) | 2.40 | 2.00 | 0.67 | 1.00 | 4.00 |
| Age (yrs) | 55.10 | 55.00 | 13.80 | 24.00 | 90.00 |
| Wealth (x €1000) | 270.28 | | 240.99 | | |
| Gross income (x €1000) | 39.81 | | 27.12 | | |
| Male (%) | 76.30 | | | | |
| Higher vocational or university education (%) | 52.30 | | | | |
| Employee (%) | 50.60 | | | | |
| Partner (%) | 74.30 | | | | |
| Households with kids (%) | 28.40 | | | | |
| N | 257 | | | | |

Note: This table provides descriptive statistics for the full DHS sample, the subsample of all investors, and the subsample of investors for which detailed portfolios could be retrieved.

The average return loss of our sample is 57 bps. The mean investor thus could have earned 0.57 percentage points more in annual return than they did, given the risk of their portfolio. Clearly, the distribution is right-skewed, since the median return loss is just 28 bps, which also shows that the median household performs relatively well. The minimum return loss is only about 1 bps, while the maximum return loss equals a huge 877 bps.

Our financial literacy factor ranges between −2.54 and +0.93. To clarify these scores[3], we present the average number of correct answers (out of 11) in Table 1. The average respondent in the full DHS sample answers 6.4 questions correct, while this is 8.6 for our investor sample. Only about 7% of the people correctly answer all questions in the full sample, while this is 20% for our sample. Clearly, investors tend to score better on average.

The mean self-assessed financial literacy score in our sample is 2.4, which is slightly higher than the mean in the whole sample (2.15). Combining this perceived literacy score with the measured score; we note that our sample is a bit more overconfident than the whole sample on average (0.19 versus 0.00 for the full sample). In addition, the distribution is right-skewed, as shown by the medians being lower than the means (−0.28 and −0.25, respectively).

In our sample, most people indicate "professional financial adviser" (24.1%) as their main source of advice, followed by "parents, friends or acquaintances", and "financial information from the Internet". This number is not so different from the 27.1% financial advice seekers in the whole sample.

### 2.4. Model and Methods

We will thus test whether overconfidence relates to the return loss, and whether hiring a financial advisor mediates this relationship. We present a graphical representation of our proposed relationships in Figure 1.

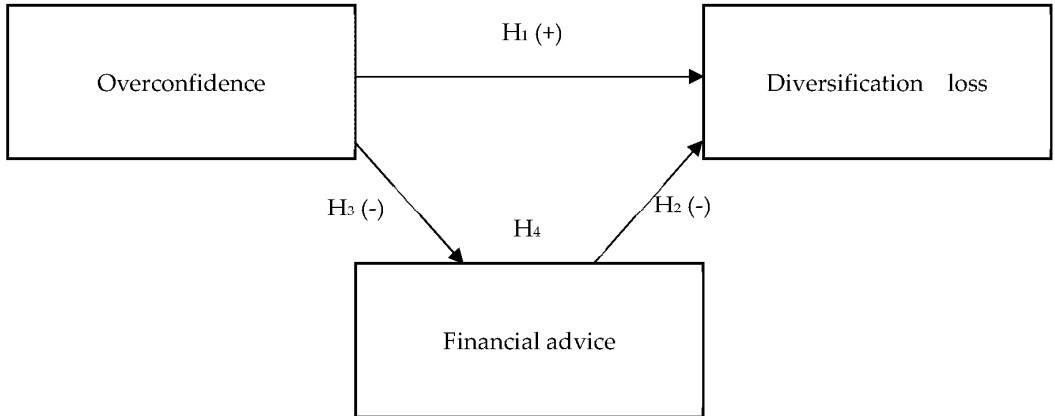

**Figure 1.** Research Framework.

To test the effect of overconfidence on financial advice (H3), we estimate a linear probability model[4], the other models are estimated using standard OLS. To test whether financial advice mediates the relationship between overconfidence and return loss two approaches are followed.

The first approach is the methodology put forward by Baron and Kenny (1986). We first test the relationship between overconfidence and the return loss, and second between overconfidence and financial advice. Third, we estimate whether the effect of overconfidence on the return loss changes if financial advice is included as an additional control variable in the first estimation. If the relationship between overconfidence and return loss becomes insignificant in this step, there is full mediation: overconfidence relates to the return loss, but only through its relationship with financial advice. If the relationship becomes weaker, there is so-called partial mediation: overconfidence directly relates to the

return loss, but also indirectly via financial advice. Furthermore, financial advice should show a significant relationship with the return loss in this third estimate. We calculate the degree of mediation as the ratio between the indirect effect and the total effect.

Yet, the Baron and Kenny (1986) approach has been criticized for different reasons (Hayes 2009). For example, it appears that the methodology has relatively low power. Furthermore, it does not immediately test the significance of the indirect effect. Therefore, we test for the significance of the indirect effect (w) by using a bootstrapping procedure (Hayes 2009). The idea is to resample with replacement from the original sample k times to come up with a distribution of the indirect effect. The corresponding confidence interval shows whether the effect is significant or not.

## 3. Results

### 3.1. Univariate Results

To provide a first feeling of the main relationships between our variables, we present some univariate statistics in Table 2. Panel A relates the return loss to overconfidence quartiles. Clearly, the return loss rises steadily over the first three quartiles (from 46 bps to 54 bps), after which it increases sharply. It averages at 77 bps in the top overconfidence quartile, which is significantly higher than the lowest quartile. This supports the hypothesis that overconfident people tend to incur higher losses from under-diversification.

**Table 2.** Return Loss, Univariate Statistics.

| Panel A. Overconfidence | | Panel B. Financial Advice | |
|---|---|---|---|
| Q1 (low) | 0.46 (0.06) | Professional financial adviser | 0.38 (0.06) |
| Q2 | 0.51 (0.11) | Other type of advice | 0.62 (0.08) |
| Q3 | 0.54 (0.13) | | |
| Q4 (high) | 0.77 (0.17) | | |
| Difference (Q4–Q1) | 0.31 * (0.18) | Difference | −0.24 * (0.14) |

Note: This table presents the return loss by overconfidence quartiles (Panel A) and by having a professional financial advice or not (Panel B). We present the means and the corresponding standard errors (in parentheses). * denotes significance at $p < 0.05$.

Similarly, Panel B of Table 2 reports the relationship between the return loss and financial advice seeking. In line with our second hypothesis, people that rely on a professional financial adviser earn a return loss that is on average 24 bps lower than people who rely on other types of advice. At least in this univariate setting, advised investors thus achieve better portfolio results.

### 3.2. Multivariate Results

Our main analysis relates overconfidence to return losses, while controlling for the many factors that may drive investment performance as well, such as age, gender and wealth. In line with Hypothesis 1, the results in Table 3 (Column 1) show that overconfidence is significantly and positively related to the return loss. A higher degree of overconfidence, thus, results in poorer investment outcomes. More specifically, a one standard deviation increase in overconfidence, ceteris paribus, corresponds to a 15-bps increase in the return loss, a considerable effect, considering the average return loss of 57 bps. Moving from the lowest to the highest score on our overconfidence measure, results in a negative impact of more than 700 bps in annual return. The finding implies that more overconfident people tend to incur higher losses from under-diversification.

**Table 3.** Return Loss, Overconfidence and Financial Advice Seeking, Multivariate results.

|  | 1 | 2 |
|---|---|---|
| Overconfidence | 0.15 * <br> (0.06) | 0.14 * <br> (0.06) |
| Financial advice |  | −0.20 * <br> (0.10) |
| Age | 0.02 * <br> (0.01) | 0.02 * <br> (0.01) |
| Male | 0.13 <br> (0.15) | 0.10 <br> (0.14) |
| Education |  |  |
| Primary or pre-vocational education | 0.07 <br> (0.13) | 0.06 <br> (0.13) |
| Senior vocational education | 0.25 <br> (0.18) | 0.26 <br> (0.18) |
| Pre-university education | (0.24) <br> (0.18) | (0.25) <br> (0.18) |
| Log gross income | −0.06 <br> (0.03) | −0.06 <br> (0.03) |
| Retired | (0.17) <br> (0.30) | (0.15) <br> (0.31) |
| Self-employed | 1.38 <br> (0.93) | 1.39 <br> (0.93) |
| Employee | 0.01 <br> (0.29) | 0.02 <br> (0.29) |
| Partner | −0.68 ** <br> (0.20) | −0.69 ** <br> (0.20) |
| Kids | 0.15 <br> (0.12) | 0.16 <br> (0.12) |
| Log wealth | 0.01 <br> (0.02) | 0.02 <br> (0.02) |
| Constant | 0.26 <br> (0.41) | 0.24 <br> (0.41) |
| No. of observations | 255 | 255 |
| $R^2$ | 0.20 | 0.21 |

Note: This table presents estimation results of overconfidence and financial advice seeking on return losses. We present robust standard errors in parentheses. *, and **, denote significance at $p < 0.05$, and $p < 0.01$, respectively.

Table 4 shows that overconfidence is significantly negatively related with financial advice, in line with our fourth hypothesis. Specifically, a one standard deviation increase in overconfidence corresponds on average to a four-percentage point reduction in the propensity to seek advice. Moving from individuals that score highest in terms of overconfidence, they exhibit a 20-percentage point reduction in advice seeking compared to the lowest. Looking at the control variables, wealth seems to matter most. A 10 percent increase in wealth corresponds to an increase in the propensity to seek financial advice by approximately 0.3 percentage points.

**Table 4.** Overconfidence and Financial Advice Seeking.

|  | 1 |
|---|---|
| Overconfidence | −0.04 * |
|  | (0.02) |
| Age | 0.00 |
|  | 0.00 |
| Gender | (0.09) |
|  | (0.06) |
| Education |  |
| Primary or pre-vocational education | (0.03) |
|  | (0.07) |
| Senior vocational education | 0.05 |
|  | (0.07) |
| Pre-university education | 0.01 |
|  | (0.08) |
| Log gross income | (0.02) |
|  | (0.02) |
| Retired | 0.08 |
|  | (0.10) |
| Self-employed | (0.01) |
|  | (0.16) |
| Employee | 0.02 |
|  | (0.09) |
| Partner | (0.04) |
|  | (0.06) |
| Kids | 0.14 * |
|  | (0.07) |
| Log wealth | 0.03 ** |
|  | (0.01) |
| Constant | 0.23 |
|  | (0.21) |
| No. of observations | 255 |
| $R^2$ | 0.06 |

Note: This table presents estimation results of overconfidence on professional financial advice seeking using a linear probability model. We present robust standard errors in parentheses. *, and **, denote significance at $p < 0.05$, and $p < 0.01$, respectively.

Now we established that overconfidence relates both to a higher loss from poor portfolio diversification, and a lower propensity to ask for expert help, we can now test whether the link between overconfidence and return loss is mediated by financial advice. We present our results in Column 2 of Table 3.

People who obtain professional financial advice have a return loss that is on average 20 bps lower than individuals who rely on other types of advice. In line with Hypothesis 2, people that opt for expert advice, thus achieve better investment outcomes. This supports the findings of Kramer (2012).

Interestingly, the effect of overconfidence falls slightly, but remains significant when we add financial advice to the model. This indicates that there does not appear to be full mediation: there is a direct effect of overconfidence on the return loss, accompanied by an indirect effect through the effect of overconfidence on financial advice (and the effect of advice on the return loss).

When comparing the two columns of Table 3, it appears that the effect of overconfidence decreases with 1.2 bps (from 15 bps to 14 bps) when financial advice is included

in the model. Using a bootstrapping procedure based on Hayes (2009), we assess the significance of the indirect effect. Using 5000 bootstrap samples (with replacement), the bias-corrected confidence interval (see e.g., MacKinnon et al. 2004; Mallinckrodt et al. 2006; Hayes 2013) of the indirect effect shows that it is significant at the five percent level. The degree of mediation is equal to a little under 10 percent (calculated by dividing the estimated indirect effect of 1.2 bps by the total effect of 15 bps). A one standard deviation increase in overconfidence corresponds to a 15 bps higher return loss, of which the effect of overconfidence on financial advice (i.e., the indirect effect) explains roughly 1.2 bps. The rest of the total effect (roughly 13.8 bps) reflects the direct effect of overconfidence on the return loss. In other words, more overconfident people tend to have a higher return loss because of a direct effect, but also because they are less likely to rely on financial advice (which in turn corresponds to a higher return loss as well). This result is in line with Hypothesis 4.

Our control variables indicate that age is positively related to the return loss. This finding supports Korniotis and Kumar (2011) who find that older investors underperform because of cognitive aging. Gross income appears negatively related, in line with the view that investors that are more sophisticated make better decisions (Bailey et al. 2008). Interestingly, people with a partner in the household tend to earn a lower return loss on average. This appears to be in line with Barber and Odean (2001), who find that, particularly for men, being married can help in achieving better performance, since the partner may influence the decisions made, and since women tend to be less overconfident than men. In fact, if we split the current sample on gender, the result appears to be entirely driven by the subsample of men.

## 4. Robustness Tests

We perform various additional analyses to test for the robustness of our main findings. First, one would expect that if overconfident investors incur higher losses from under-diversification, they are less efficient risk-takers. This would be signified by a higher degree of avoidable risk in their portfolios. We, therefore, test for the relationship between overconfidence and financial advice on total, systematic, and idiosyncratic risk of the portfolio. Total risk refers to the volatility of the expected portfolio returns, systematic risk to the portfolio beta, and idiosyncratic risk to the part of the volatility not explained by the portfolio beta. The results in Table 5 support the view that overconfident investors tend to have riskier portfolios (Odean 1998). More importantly, this result is driven both by more efficient risk-taking (reflected in a higher systematic risk) and by more uncompensated risk-taking (reflected in a higher idiosyncratic risk). Furthermore, relying on a professional financial adviser relates negatively to systematic as well as idiosyncratic risk, which is in line with Kramer (2012).

**Table 5.** Portfolio risk, overconfidence, and financial advice.

|  | 1 | 2 | 3 |
|---|---|---|---|
| Overconfidence | 0.02 * (0.01) | 0.01 * 0.00 | 0.01 * (0.01) |
| Financial advice | −0.04 * (0.02) | (0.01) (0.01) | −0.04 ** (0.01) |
| Controls included (see Table 3) | Yes | Yes | Yes |
| No. of observations | 255 | 255 | 255 |
| $R^2$ | 0.07 | 0.05 | 0.09 |

Note: This table provides estimation results of overconfidence and financial advice seeking on various risk measures. The dependent variables are total risk (Model 1), systematic risk (Model 2), and idiosyncratic risk (Model 3). In parentheses, we present robust standard errors. *, and **, denote significance at $p < 0.05$, and $p < 0.01$, respectively.

Second, given the prominence of overconfidence as our explanatory variable, we apply two alternative proxies. Specifically, we use data based on two questions eliciting the respondents' expected inflation estimates. We derive a confidence interval by subtracting their low inflation estimate from their high one. We then create a dummy that equals one for people whose highest and lowest inflation estimates differ by less than one percentage point, and zero otherwise. Although those people may actually have superior knowledge and are capable in forecasting inflation, it appears that forecasting inflation is very hard (Mankiw et al. 2003). Our dummy may therefore be interpreted to reflect overconfidence, especially given that we control for measured literacy in our estimations. Our inflation dummy does not appear to be significantly related to the propensity to seek financial advice for both samples, yet it shows a significantly positive relationship with the return loss, thus providing partial support for our key findings.

We also apply a method similar to the one used by Gentile et al. (2016). We create two dummies: the first dummy (called high self-assessment dummy, HSA) equals one if people rate themselves above the sample average, and the second dummy (called high financial literacy dummy, HFL) equals one if people score above the sample median with respect to financial literacy. Then we use the difference between HSA and HFL as our alternative overconfidence proxy (the resulting outcomes are −1 for underconfident people, 0 for well-calibrated people, and 1 for overconfident people). In our re-estimations this new variable relates significantly negatively to advice seeking, and significantly positively related to the return loss, supporting our initial findings.

As a third robustness test, we check whether our results hold also for a related concept such as excessive optimism. To proxy for excessive optimism, the following question is used: "How likely is it that you will attain (at least) the age of 80?"[5] A dummy is created that equals one if respondents think they have a chance of ten (on a scale from one to ten) to reach that age, and zero otherwise. Furthermore, it might be that those people are correct in their estimate because of good health. Therefore, we include the body mass index as a proxy for health in the estimation. We find that our proxy for excessive optimism relates negatively to the propensity to hire professional financial advice. The effect of excessive optimism on return loss is in the expected direction, although it does not reach significance at conventional levels.

Fourth, to lower the possibility that omitted variable bias drives the results, we add the following additional controls to our specifications: cognitive ability, risk aversion, and experience, that all may act as additional drivers of proper diversification. In our main estimations, we initially excluded them to prevent a large reduction of our sample size. Cognitive abilities might be related to both financial advice seeking and the return loss. Higher cognitive abilities make the complex task of processing information related to investing easier, leading to better investment outcomes (Grinblatt et al. 2012). Higher cognitive ability may also make people more aware of their limitations (Kruger and Dunning 1999) and may hence affect their perceived need for expert assistance. To proxy for cognitive abilities, we carry out a factor analysis on five basic numeracy questions of the financial literacy module (see Van Rooij et al. 2011 for details). The factor analysis is appropriate according to Bartlett's test of sphericity ($p < 0.01$) and the KMO measure of sampling adequacy (KMO = 0.85).

Furthermore, it might be that omitting people's risk aversion drives our results: risk averse people have been found to be less overconfident (Eckel and Grossman 2008), and more likely to rely on a financial adviser (Calcagno and Monticone 2015). We operationalize risk aversion by performing a factor analysis on six questions relating to risk aversion[6]. The factor analysis is appropriate according to Bartlett's test ($p < 0.01$) and the KMO measure of sampling adequacy (KMO = 0.67).

Lastly, we add investment experience to all specifications. More experienced investors may consider themselves to be in less need of seeking professional financial advice, because they have obtained more knowledge with regards to the investing process. Likewise, more experienced investors may invest more efficiently, because they learned from their previous

investment strategies. We operationalize investment experience in two different ways. Firstly, the number of years are counted in which a certain household is included in the CentER data panel (before our sample period) while holding risky assets. Secondly, to account for the fact that households are likely to be included in the panel for a limited time, the number of years in which a household holds risky assets is divided by the total number of years that the household has been observed in the panel before our sample period.

Most importantly, our key results remain unaltered after the inclusion of these three additional control variables. In fact, our results become even stronger: financial advice relates now significantly to the return loss at the one percent level, and the degree of mediation increases to roughly 15%. Furthermore, all additional controls are negatively related to the return loss (and marginally significant): people with higher cognitive abilities, that are more risk averse, or have more investment experience tend to incur lower return losses.

As a fifth robustness test, in line with Von Gaudecker (2015), we re-estimated all equations after subtracting mutual fund fees from the gross portfolio returns. Advisors typically have an incentive to advice more expensive (that is higher fee) mutual funds, reducing their positive impact on the returns. Overconfident investors may also select active (that is: more expensive) funds in their believe to be able to beat the market. Subtracting fees, however has negligible effects on our results.

Sixth, apart from omitted variable bias as discussed before, endogeneity may arise from other sources. Reversed causality for example implies that investment performance may actually influence overconfidence, and advice seeking. The use of a good instrument would solve this issue, but unfortunately, no such instrument is available in the data. Yet, Barber and Odean (2001) argue that overconfidence is more likely in men than in women. Therefore, we perform a sample split on gender, and re-estimate all our equations. These estimates show that our findings are mainly driven by men for both the propensity to seek financial advice and the return loss. This provides some support for the causality argument (since gender is clearly exogenous). We do note, however, that this approach does not fully rule out the possibility that our results suffer from endogeneity.

## 5. Conclusions and Discussion

The literature on household finance shows that households tend to incur losses from investment mistakes. Given the mounting evidence that psychological biases may help to explain this finding, we test for the role of overconfidence on diversification losses, and the role of financial advice therein.

We show that most people appear to incur modest losses from under-diversification: the median household incurs a return loss of 28 bps a year. Yet, overconfident investors incur significantly higher losses in two distinct ways. First, we find a direct relationship between overconfidence and losses from under-diversification. This supports the view that holding a focused (thus undiversified) portfolio is a sensible strategy in the eyes of an overconfident investor because both portfolio risk is underestimated, and stock-picking ability is overestimated. Second, overconfident investors see less need to ask for expert advice, which, at least when applied to diversification quality, hurts their performance as well. We support these findings by various robustness tests. This paper thus complements the notion put forward by Bhattacharya et al. (2011) that investors who may benefit most from financial advice (i.e., those who incur high return losses, being overconfident people) are least likely to actually do this.

We cannot infer causality given the cross-sectional nature of our data, so we must take care not to over-interpret based on our results. For example, it could be that both overconfidence and return losses are driven by an omitted factor such as intelligence. Financial advice seeking may be driven by financial outcomes, rather than the other way around. Future research should attempt to overcome our concerns in order to identify causal connections. We do note, however, that our findings closely match those of a related experimental study by Hung and Yoong (2013) that aimed specifically at identifying casual

effects of financial advice seeking on portfolio allocation. Just as in our study, their group of subjects that choose deliberately to turn to advice did significantly better that the group that choose not to. Our study suggests that overconfidence could be one of the mechanisms behind their findings.

Albeit these limitations, our results aid in the current discussion among policymakers on the role of financial advice in financial decision-making. Some argue that stimulating the uptake of financial advice may well be a worthwhile strategy to explore. In fact, the introduction of MiFID II legislation actually tries to stimulate the uptake of financial advice, by limiting the number of financial products available to self-deciders. Our results imply that this discussion may benefit from also taking biased self-attributes into consideration. First, it may hinder the uptake of advice and second, it may well be a problem of greater magnitude than limited advice seeking itself.

**Author Contributions:** Conceptualization, S.P.M.B. and M.M.K.; methodology, S.P.M.B.; software, S.P.M.B.; validation, S.P.M.B.; formal analysis, S.P.M.B.; investigation, S.P.M.B.; resources, S.P.M.B.; data curation, S.P.M.B.; writing—original draft preparation, S.P.M.B.; writing—review and editing, M.M.K.; visualization, S.P.M.B.; supervision, M.M.K.; project administration, M.M.K.; funding acquisition, not applicable. All authors have read and agreed to the published version of the manuscript.

**Funding:** This research received no external funding.

**Institutional Review Board Statement:** Ethical review and approval were waived for this study, due to the use of a publicly available dataset that was set up for academic research purposes and for which all panel participants gave their consent to Centerdata (https://www.centerdata.nl/, 7 April 2016) before.

**Informed Consent Statement:** Informed consent was obtained from all subjects involved in the study.

**Data Availability Statement:** We used the 2005 Wave of the Dutch Household Survey that is publicly available for researchers at https://www.dhsdata.nl/site/users/login (7 April 2016).

**Conflicts of Interest:** The authors declare no conflict of interest.

## Notes

1 $\left(\frac{S_m - S_h}{S_h}\right)$ measures the diversification loss, in line with Von Gaudecker (2015). Furthermore, to limit the influence of outliers, the return loss is winsorized at the 99th percentile. Robustness is checked for different percentiles.cked for different percentiles. d for different percentiles. loss is winsorized at the 9th ition, since the subsample of detaile.

2 There appear to be relatively many missing values for gross income (201 out of 1277). To increase the sample size, these missing values are imputed by the predicted values from a regression on observable characteristics that are non-missing for the full sample such as age, employment, and education. The results are robust to excluding the observations with missing values for gross income from the estimations, unless otherwise indicated.

3 The underlying statistics are available upon request.

4 Clearly, a logit or probit type of model would be more suitable for this situation. Yet, mediation analysis requires the different models to be of the same type, which implies that a linear probability model is the best option in this case. The results are robust to using a logit or probit model.

5 This is a standard question in the DHS panel. The value of 80 years was thus outside the authors' control.

6 Underlying questions and statistics available upon request.

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
