# Peer review of "Overconfidence, Financial Advice Seeking and Household Portfolio Under-Diversification"

_jrfm, doi:10.3390/jrfm14110553_

Round 1
Reviewer 1 Report
General issues
The paper transmits a clear idea and can be a useful contribution for decision-makers. Its approach is innovative. The methodology is appropriate for the purpose of the paper. The paper is well structured. I have some minor suggestions, listed below.
- On page 4, rows 159-162, I think that this idea should be explained deeper: “Specifically, we create a dummy which equals one for people whose highest and lowest inflation estimates differ by less than one percentage point, and zero otherwise. Although narrow confidence intervals may represent superior capability in inflation forecasting, Mankiw et al., (2003) shows it may actually reflect overconfidence.”. For instance, have you checked how these estimations have fitted realized inflation?
- On page 11, row 371 - see “How likely is it that you will attain (at least) the age of 80?”, please, explain how have you chosen the value of “80”.
- Even the article is well written (intelligible for the reader), I suggest to the authors to use an equation editor (see Eq.1 on page 3).
- In the same line, the text should be carefully re-read (e.g., on page 3, row 139, replace the present text “β_hrefers to the household beta” with “β_h refers to the household beta”).
Conclusion:
In my opinion, the article can be published after considering for these minor suggestions.
Author Response
Dear reviewer,
Thanks for your valuable suggestions. We addressed them as follows:
- We better explained why a narrow inflation interval proxies for overconfidence and checked how the respondents' expectations fitted the realized inflation. We confirmed that this robustness test proxies reasonable for overconfidence. The group that we labelled as overconfident based on this measure, was substantially more often surprised that the non-overconfident group.
- We explain how we attained at the age of 80 in the question on live expectancy. This number is a standard question in the DHS survey, and was thus not deliberately chosen by us. We still believe it to be an acceptable proxy for excessive optimism.
- We used an equation editor.
- We improved some typos.
Kind regards, Marc Kramer & Stijn Broekema
Reviewer 2 Report
The abstract should be supplemented with the research methods used. In addition, it should at least indicate the size of the research sample and the year of the study.
Author Response
Dear reviewer,
Thank you for your valuable suggestion. We revised our abstract and now include the sample size, the sample period and a brief explanation of our methodology.
Wit kind regards,
Marc Kramer & Stijn Broekema